# Chest computed tomography outcomes in a randomized clinical trial in cystic fibrosis: Lessons learned from the first ataluren phase 3 study

Harm A. W. M. Tiddens[1,2]*, Eleni-Rosalina Andrinopoulou[3], J. Stuart Elborn[5], Eitan Kerem[6], Nynke Bouma[1], Jochem Bosch[1], Mariette Kemner-van de Corput[1,2]

1 Department of Pediatric Pulmonology and Allergology, Erasmus Medical Center Sophia Children's Hospital, Rotterdam, The Netherlands, 2 Department and Radiology and Nuclear Medicine, Erasmus Medical Center, Rotterdam, The Netherlands, 3 Department of Biostatistics, Erasmus Medical Center, Rotterdam, The Netherlands, 4 Aruvant Biotech, New York, NY, United States of America, 5 Centre for Experimental Medicine, Queen's University Belfast, Belfast, United Kingdom, 6 Department of Pediatrics and CF Center, Hadassah-Hebrew University Medical Center, Jerusalem, Israel

* h.tiddens@erasmusmc.nl

**Data Availability Statement:** All relevant data are within the manuscript and its Supporting Information files. Annotated CTs by LungAnalysis

## Abstract

A phase 3 randomized double blind controlled, trial in 238 people with cystic fibrosis (CF) and at least one nonsense mutation (nmCF) investigated the effect of ataluren on $FEV_1$. The study was of 48 weeks duration and failed to meet its primary endpoint. Unexpectedly, while $FEV_1$ declined, chest computed tomography (CT) scores using the Brody-II score as secondary outcome measures did not show progression in the placebo group. Based on this observation it was concluded that the role of CT scans in CF randomized clinical trials was limited. However, more sensitive scoring systems were developed over the last decade warranting a reanalysis of this unique dataset. The aim of our study was to reanalyse all chest CT scans, obtained in the ataluren phase 3 study, using 2 independent scoring systems to characterize structural lung disease in this cohort and to compare progression of structural lung disease over the 48 weeks between treatment arms. 391 study CT scans from 210 patients were reanalysed in random order by 2 independent observers using the CF-CT and Perth-Rotterdam Annotated Grid Morphometric Analysis for CF (PRAGMA-CF) scoring systems. CF-CT and PRAGMA-CF subscores were expressed as %maximal score and %total lung volume, respectively. PRAGMA-CF subscores %Disease (p = 0.008) and %Mucus Plugging (p = 0.029) progressed over 48 weeks. CF-CT subscores did not show progression. There was no difference in progression of structural lung disease between treatment arm and placebo independent of tobramycin use. PRAGMA-CF Chest CT scores can be used as an outcome measure to study the effect of potential disease modifying drugs in CF on lung structure.

are stored on ErasmusMC servers and can be reviewed within the ErasmusMC IT environment upon request. The raw data (Original CTs and annotated CTs) cannot be shared by us because the CTs are owned by a 3rd party (PTC pharmaceuticals Inc.). Furthermore, depending on the question written permission by PTC for access to the CTs might be needed as PTC pharmaceuticals is the owner of the CT dataset. Results of the CF-CT and PRAGMA-CF analysis are available as a locked spreadsheet as part of the Supporting Information files. Data access requests can be sent to unganalysis@erasmusmc.nl (email address for the lung analysis imaging analysis laboratory) and CFTR. sophiaresearch@erasmusmc.nl (email address for all research coordinators involved in all CF related research).

**Funding:** This investigator initiated study was funded through an unconditional grant by PTC Pharmaceutics, Inc. The corresponding author was given full access to all data that was requested to PTC. Only data within the scope of the study were requested. The corresponding author had final responsibility for the decision to submit for publication. The funder provided support in the form of salaries for EA, NB, JB, and MKvdC. Funder had no role in the image analysis study design and analysis, decision to publish.

**Competing interests:** JM worked at the time of the manuscript preparation at PTC and owns stock of PTC, he supplied us with relevant data when requested and he critically read the manuscript and gave relevant input where needed fulfilling all requirements for co-authorship. EK received a research grant through his hospital to perform the ataluren Phase III study and acted as Medical & Scientific Consulting Board member and received reimbursement for his time and for travel expenses. HT is director of the Erasmus MC LungAnalysis laboratory, he is a co-inventor of PRAGMACF which is patented. All financial aspects of the patent are handled by the Erasmus MC.

## Introduction

Outcome measures derived from chest computerized tomography (CT) are considered a more sensitive alternative for forced expiratory volume in 1 second ($FEV_1$) to monitor structural lung changes [1–3]. To date only few large randomized controlled trials (RCTs) in cystic fibrosis (CF) have been conducted that included chest CT as an outcome measure [4, 5].

One of these studies was a RCT comparing ataluren to placebo, which was conducted from 2009 into 2011 [5]. This was a randomized, double-blind, placebo-controlled phase 3 study that included 238 CF patients, 6 years and older with at least one nonsense mutation (nm). The primary endpoint of this study was relative change in % predicted $FEV_1$ from baseline to week 48 assessed by spirometry; the secondary endpoint was the rate of pulmonary exacerbations. The remainder of the endpoints presented were tertiary or exploratory. Change in lung CT scores were included as one of the exploratory outcome measures. The Brody-II CT scores were used as image analysis method [6]. The results of this study showed that there were no statistical significant differences between treatment arms in the primary outcome measure ($FEV_1$) and secondary and tertiary outcomes including CT scores. In both the active and placebo groups %predicted $FEV_1$ declined from baseline over the 48 weeks study duration. However, a post hoc analysis of the subgroup of patients not using chronic inhaled tobramycin showed a significant 5–7% difference in relative change from baseline in % predicted $FEV_1$ and fewer exacerbations in ataluren treated patients relative to the placebo group at week 48 [5]. Based on this post hoc analysis it was concluded that treatment with ataluren might be beneficial for nmCF patients not receiving chronic inhaled tobramycin as tobramycin may interfere with the ribosomal binding mechanism of ataluren. For this reason, a second phase 3 study was conducted including only nmCF patients not treated with inhaled tobramycin using % predicted $FEV_1$ as primary endpoint. In this subsequent study chest CT scans were no longer included as outcome measure (clinicaltrials.gov: NCT02139306).

It was surprising that despite the large number of CF patients with severe mutations in the first phase 3 ataluren study there was no disease progression over 48 weeks study duration as determined by the Brody–II CT scores. The authors of the study concluded in the online supplement that 'the role of chest CT scans in study designs for CF clinical trials was limited'. This finding is in contrast to multiple smaller observational cohort studies, which have consistently demonstrated progression of CT scores over one- to two-year intervals and that CT scores are more sensitive than $FEV_1$ as a marker of progression of structural lung disease [1, 3]. Unfortunately, details on the image analysis of this unique set of CT scans have not been published.

Since the completion and analysis of the first phase 3 ataluren RCT, two potential more sensitive and better standardized scoring systems for chest CT scans have been developed. The first is the CF-CT scoring system, which is an upgraded version of the Brody-II scoring system used for scoring of the CT scans in the first phase 3 ataluren study. The CF-CT scoring system uses well standardized definitions, reference images and standardized training of observers [7]. The second system is Perth-Rotterdam Annotated Grid Morphometric Analysis for CF (PRAGMA-CF) which is based on a morphometric approach and which has been shown to be more sensitive for the quantification of early CF lung disease relative to the Brody-II scoring and CF-CT scoring system [8]. Based on the availability of these two improved CT scoring systems and importance of the unique large ataluren chest CT scan dataset for CF drug studies, we repeated the CT scan analysis.

The aim of our study was to reanalyse all first phase 3 ataluren chest CT scans using both the CF-CT and PRAGMA-CF as independent scoring systems to phenotype structural lung disease in nmCF patients and to compare disease progression of structural lung disease over

the 48 weeks between treatment arms. We also did a post hoc analysis to investigate whether our more sensitive CT scan analysis would have given useful information to the effectiveness of ataluren in nmCF patients with or without tobramycin treatment. We hypothesized that significant progression of structural lung disease can be observed in the study cohort using CF-CT and PRAGMA-CF scoring systems.

## Methods

### Study population

The randomised, double-blind, placebo-controlled, phase 3 trial of ataluren was performed between August 2009 and November 2011 at 36 sites in 11 countries in North-America and Europe [5]. Most important inclusion criteria included: age ≥6 years; abnormal nasal potential difference; sweat chloride >40 mmol/L; documentation of the presence of a nonsense mutation in at least one allele of the CF Transmembrane Conductance Regulator gene [5]. Inclusion and exclusion criteria of the ataluren study are detailed in the 1.1 in S1 File.

The protocol of the phase 3 ataluren trial (ClinicalTrials.gov, number NCT00803205) was reviewed and approved by the ethics committee or institutional review board of each participating institution. Written informed consent was obtained from patients or their custodians prior to patient screening. External oversight of the study was provided by a Study Steering Committee and an independent Data Monitoring Committee. The reanalysis of CTs was executed on coded CTs.

### Chest CT scan

An inspiratory spiral low dose chest CT scan was obtained at the start of study (SOS) and end of study (EOS) in the ataluren phase 3 study in 210 of the 238 patients with nmCF. CT scanner specifics can be found in S1 Table in S1 File. The reason why 28 patients did not contribute a SOS and EOS CT was not defined in the paper by Kerem et al. and this information was not available in the data files supplied to us by PTC.

### CT Scoring

We received 420 de-identified SOS and EOS CT scans from PTC of which 391 CT scans were of sufficient quality to be scored in random order using the CF-CT [7] and PRAGMA-CF [8] scoring systems. Each scoring system was performed by a single certified observer and both had 2 years of image analysis experience. Details on the training of the observers are given in 1.2 in S1 File.

The CF-CT scoring system evaluates the five lung lobes and lingua for severity and extent of central and peripheral CT abnormalities through eye balling. The following sub-scores were assessed: bronchiectasis, airway wall thickness (AWT), mucus plugging, and parenchymal disease. Each subscore is scored for each lobe on the inspiratory scan as absent, occupying 0–33%, 33–66%, or >66% of the lobe, and next multipliers are applied. Resulting scores range from 0 to 72, 0 to 54, 0 to 36, and 0 to 54 for these 4 components, respectively. The maximal possible total CF-CT score summing these subscores is 216 points. For statistical analysis CF-CT subscores are expressed as % of maximal score. The CF-CT scoring system takes around 30 minutes per CT scan.

PRAGMA-CF is a morphometric system that computes the volume fraction of structural lung components using a grid overlaying 10 equally spaced axial CT slices between lung apex and base. Each grid cell is scored with the following hierarchical system (highest to lowest priority): bronchiectasis, mucus plugging, AWT, atelectasis, and normal lung structure.

PRAGMA-CF subscores are expressed as % total lung volume. The composite score %Disease for CF-CT and for PRAGMA-CF was computed, which is defined as the sum of %Bronchiectasis, %Mucus Plugging, and %AWT. To compute changes (Δ) in subscores over the study period, SOS values were subtracted from EOS values of patients with two available CT scans. The PRAGMA-CF scoring system takes between 30–40 minutes per CT scan.

Regions of low attenuation were not assessed as most (95%) patients had only three expiratory slices available [9]. To calculate intra-observer variability, a random subset of 20 CT scans was rescored after a one-month interval by both observers.

## Statistics

Continuous variables are presented as mean and standard deviation (SD).

Intra-observer reliability for CT scoring was assessed using the intra-class correlation coefficient (ICC), a two-way mixed-effects model. Because no universally applicable standards are available for reliability, an ICC between 0 and 0.4 was considered poor, between 0.4 and 0.6 moderate, between 0.6 and 0.8 good and greater than 0.8 excellent.

We performed regression models and obtained the Pearson correlation (r) to investigate the association between PRAGMA-CF %Bronchiectasis with CF-CT %Bronchiectasis, PRAGMA-CF %Mucus Plugging with CF-CT %Mucus Plugging, and between PRAGMA-CF %Disease with CF-CT %Disease separately at SOS and EOS.

Bland Altman plots were generated plotting the difference between PRAGMA-CF and CF-CT subscores versus their mean values at both SOS and EOS. This was done for %Bronchiectasis, %Disease, and %Mucus Plugging being the subscores showing a significant change from SOS to EOS. Since the scale of the two methods is different we first standardized the variables.

Linear mixed effects models [10] were used to assess progression and compare effects of treatment for the following CT outcomes between SOS and EOS: PRAGMA-CF %Bronchiectasis, PRAGMA-CF %Mucus Plugging, PRAGMA-CF %AWT, PRAGMA-CF %Disease, CF-CT %Bronchiectasis, CF-CT %Mucus Plugging, CF-CT %AWT, CF-CT %Disease, and $FEV_1$%predicted. Confounders included in the mixed-effects models were: Time: SOS/EOS, tobramycin treatment: yes/no and treatment group: ataluren/placebo. These models take into account the unbalanced nature of the data and that measurements from the same patients may be more correlated than measurements from different patients. In particular, they consist of two parts, namely the fixed part and the random part. The fixed-effects part describes the average evolution in time of a specific clinical parameter under study (e.g. PRAGMA-CF %Bronchiectasis), where this average is taken overall from the subjects in the sample at hand and is an estimate of the evolution of the clinical parameter in the target population. The random-effects (patient specific) part describes the evolution in time for each of the patients under study. This part accounts for the correlation in the data within patients.

For some outcomes when performing a mixed-effects and regression analysis we used transformations because the assumption that the variance of the error terms is constant (homoscedasticity) was not satisfied in the original scale. In particular, the square root transformation was used for the outcomes: PRAGMA-CF %Bronchiectasis, PRAGMA-CF %Mucus Plugging, PRAGMA-CF %AWT, and PRAGMA-CF %Disease in the mixed-effects models. Furthermore, the same transformation was performed for the %Bronchiectasis, %Mucus Plugging, and %Disease in the regression analysis for the comparison between PRAGMA-CF and CF-CT in the regression models.

We finally performed a simulation study using our PTC analysis results to estimate for future intervention studies the power assuming different scenarios. We performed these

simulations for %Disease, and for %Mucus Plugging being potential reversible components of CF lung disease to obtain the power when assuming a 10, 30, and 50% reduction in progression from SOS will be obtained when treatment is initiated (more details can be found in the 1.3 and S5 Table: Power Analysis in S1 File).

No correction for multiple testing was performed. For all analysis R was used (version 4.0.2; nlme 3.1.148, lattice 0.20.41, lattice extra 0.6.29) [11]. For the ICC calculations we used IBM SPSS Statistics 21.

## Results

The study population for this blinded reanalysis of 391 CT scans of the phase 3 ataluren RCT included 210 of the 238 ataluren study patients who contributed a SOS, and/or a EOS chest CT. A flow-chart of the study profile is shown in Fig 1. 29 out of the 420 chest CT scans we received could not be scored because of poor image quality, movement artefacts, or truncated image of the lung.

CT scoring was executed in parallel to the consecutive phase 3 ataluren study. Our image analysis was completed and database locked by January 2017. In March 2017 PTC communicated the negative results of the consecutive phase 3 study performed to validate the post-hoc group analysis results in those patients not treated with inhaled tobramycin as tobramycin was hypothesized to interfere with ataluren ribosomal binding mechanism of action. In 2018 after we obtained the relevant sections of the PTC study database which allowed us to execute an in-depth image analysis including comparing study arms. Patient characteristics of the CT study population at SOS are shown in Table 1. The image analysis data for CF-CT and PRAGMA-CF scores which were used for statistical analysis are shown in S1 Data.

### CT scan analysis

The intra-ICC scores for the CF-CT scores were excellent for %Bronchiectasis, %Airway wall thickening, and %Mucus plugging and good for %Atelectasis/consolidations. The intra-ICC for the PRAGMA-CF scores were excellent for %Bronchiectasis and %Disease, good for Airway Wall Thickening, poor for atelectasis/consolidation and could not be computed for % Mucus Plugging because of too many 0 values. The exact ICCs are shown in S2 Table in S1 File. Bland Altman plots are presented in the S1A-S1C Fig in S1 File. Since the scale of the two methods is different we first standardized the variables.

### Spectrum of disease at SOS and EOS

Results of mean CF-CT and PRAGMA-CF subscores at SOS and EOS for ataluren and placebo group were merged as there was no significant difference in scores at SOS and EOS between the two groups (Table 2). Results of mean CF-CT and PRAGMA-CF subscores at SOS and EOS for the ataluren and placebo group separately are shown in S3A and S3B Table in S1 File. Disease severity varied widely between patients as is illustrated in the stacked bar plots from SOS CT scans of this patient group for the CF-CT scores (Fig 2A) and PRAGMA-CF scores (Fig 2B). The mean CF-CT %Disease score at SOS for patients on ataluren was 17.65±6.58 (Range 0 to 37%). The mean PRAGMA-CF %Disease at SOS for patients on ataluren was 9.82 ±7.45 (Range 0 to 35.5%) (see Fig 2 and Table 2).

### Correlation between CF-CT and PRAGMA-CF

Results of correlation between the mean CF-CT and PRAGMA-CF %Disease subscore at SOS and EOS CTs are shown in Fig 3. There was a clear and significant correlation for subscores

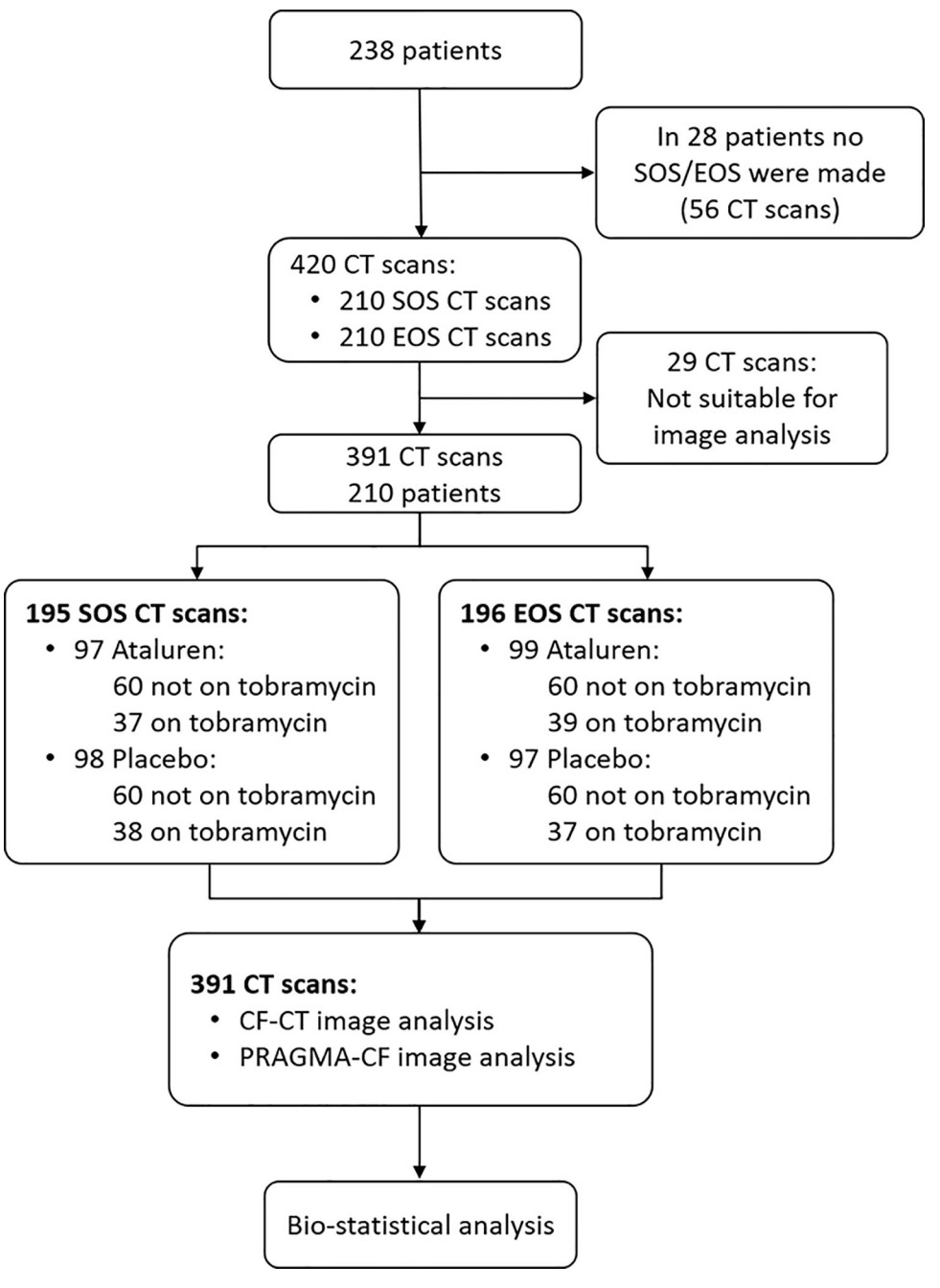

**Fig 1.**

between the two scoring systems. This was the case for SOS CT scans (Fig 3A) as well as for EOS CT scans (Fig 3B).

Correlation coefficient and p values for SOS and EOS for %Disease were (r = 0.74 and r = 0.75 respectively, both p = <0.001) (Fig 3A and 3B). Similar correlations were observed at SOS and EOS for %Bronchiectasis (r = 0.71 and r = 0.70 respectively, both p = <0.001, S2A and S2B Fig in S1 File), and for %Mucus Plugging (r = 0.55 and r = 0.49 respectively, both p = <0.001, S3A and S3B Fig in S1 File).

**Table 1. Patient characteristics at the start of study of patients with a Start of Study (SOS) and End of Study (EOS) CT which were included in the CT image analysis.**

| Patient characteristics at start of study (n = 195) | |
|---|---|
| Gender | |
| Male (%) | 49.7 |
| Female (%) | 50.3 |
| | **Mean** |
| Age (years) | 22.7 (9.7) |
| Body weight (kg) | 54.6 (13.3) |
| Body height (cm) | 162.6 (12.6) |
| Body Mass Index (kg/m$^2$) | 20.3 (3.0) |
| FEV$_1$ (l) | 2.0 (0.7) |
| FEV1 (%Predicted) | 61.3 (14.7) |

Mean (± standard deviation).

## Linear mixed models

CF-CT subscores did not show a significant progression from SOS to EOS. Progression for PRAGMA-CF subscores were found: square root of %Bronchiectasis showed a trend towards progression by 0.139 (p = 0.079) (Fig 4A). The square root of %Mucus Plugging increased significantly by 0.175 (p = 0.029), and for the square root of %Disease by 0.212 (p = 0.008) (Fig 4B). %AWT did not show a significant change.

FEV$_1$ did not show a significant change from SOS to EOS. There was no significant impact for ataluren and tobramycin treatment at baseline or over time on the linear mixed models results for %Bronchiectasis, %Mucus Plugging, %Disease and FEV$_1$.

Table 3 shows the positive results of the linear mixed model analysis for PRAGMA-CF % Mucus Plugging and %Disease. All the other linear mixed model results for CF-CT subscores and %PRAGMA-CF subscores and FEV$_1$% are shown in S4A and S4B Table in S1 File.

**Power simulation study.** Based on the PTC analysis when using PRAGMA-CF %Disease as the primary outcome measure in a new RCT 110 subjects per study arm would be needed to

**Table 2. This table shows the PRAGMA-CF and CF-CT subscores of CTs at the Start of Study (SOS) (n = 195) and End of Study (EOS) (n = 196).**

| Method | Subscore | SOS-CT | EOS-CT | delta | SD delta |
|---|---|---|---|---|---|
| CF-CT | %Bronchiectasis | 15.03±9.05 | 16.28±9.67 | 1.25 | 0.95 |
| CF-CT | %AWT | 17.81±9.39 | 18.41±8.87 | 0.6 | 0.92 |
| CF-CT | %Mucus plugging | 29.05±10.67 | 29.29±11.25 | 0.24 | 1.11 |
| CF-CT | %opacities | 14.54±5.59 | 15.04±5.37 | 0.5 | 0.55 |
| CF-CT | %Disease | 17.94±6.71 | 18.67±7.03 | 0.73 | 0.7 |
| PRAGMA-CF | %Bronchiectasis | 7.79±6.56 | 8.86±7.48 | 1.07 | 0.71 |
| PRAGMA-CF | %AWT | 0.01±0.07 | 0.02±0.07 | 0.01 | 0.01 |
| PRAGMA-CF | %ATL | 0.12±0.18 | 0.13±0.26 | 0.01 | 0.02 |
| PRAGMA-CF | %Mucus plugging | 2.14±2.47 | 2.31±2.38 | 0.17 | 0.25 |
| PRAGMA-CF | %Disease | 9.96±7.58 | 11.19±8.45 | 1.23 | 0.81 |

Results shown are mean (± standard deviation). Delta = average difference between EOS and SOS values.

For the CF-CT scoring system subscores are expressed as % of maximum score, for PRAGMA-CF as the % of total lung volume. AWT = airway wall thickness, ATL = atelectasis.

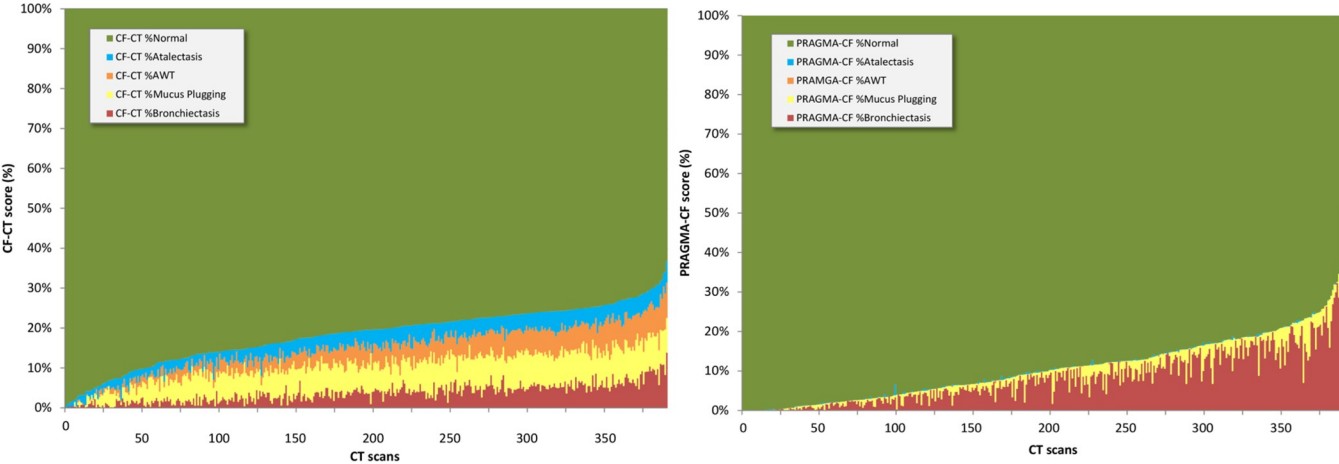

**Fig 2.** Stacked bar plots showing the wide spectrum of disease scores found by the CF-CT (A) and PRAGMA-CF scoring system (B). **A.** This stacked bar plot shows the distribution of the CF-CT %total score scores at start of study (SOS) for 195 patients. Patients are sorted based on the CF-CT %total score. %total score for each patient is subdivided by the subscores %Bronchiectasis, %Mucus Plugging, %Airway Wall Thickening and %Atelectasis. Note the wide distribution of %total score ranging from 0 to 37%. **B.** This stacked bar plot shows the distribution of the % volume of the lung at start of study (SOS) for 195 patients occupied by structural lung abnormalities as determined by PRAGMA-CF score. Patients are sorted based on the PRAGMA-CF %Disease subscore which is the sum of PRAGMA-CF %Bronchiectasis, %Mucus Plugging, and %Airway Wall Thickening. %Atelectasis (is depicted on top of %disease). Note the wide distribution of %Disease severities ranging from 0 to 35.5%.

obtain a power of 0.9 assuming an overall reduction in progression of %Disease of 50% in the active treatment arm which is generally considered a clinical relevant reduction. When using PRAGMA-CF % mucus plugging as the primary outcome measure in a new RCT 170 subjects per study arm would be needed to obtain a power of 0.9 assuming an overall reduction in progression of %mucus plugging of 50% in the active treatment arm which is generally considered a clinical relevant reduction. Other data on the simulation analysis to obtain the power when assuming a 10, 30 and 50% reduction from SOS to EOS when treatment is initiated for PRAG-MA-CF %Disease and %Mucus Plugging can be found in S5A and S5B Table in S1 File.

## Discussion

In this study we have reanalysed a unique large set of chest CT scans obtained from 210 CF patients participating in the first phase 3 ataluren study. We used two independent chest CT scoring systems to evaluate the structural changes over a 48-weeks study period in the placebo and ataluren group with and without tobramycin treatment.

The most important observation is a significant increase in severity of structural lung disease according to PRAGMA-CF %Disease and %Mucus Plugging over the 48 weeks' study period. Based on other CT scan analysis studies, progression of structural lung disease was to be expected in this cohort of CF patients with severe mutations. However, this finding is in contrast to the previous CT scan analysis performed for this study, when the Brody-II scoring system was used [5] which showed no significant changes in the Brody-II scores. There are a number of reasons to explain why we did observe disease progression and why this was not observed in the original analysis of the phase 3 RCT by PTC. The progression was observed for PRAGMA-CF subscores only and not for the CF-CT subscores. PRAGMA-CF was developed on morphometric principles which is a more sensitive scoring system especially for early lung disease in CF [12]. Our findings in this ataluren study demonstrate that PRAGMA-CF is also a more sensitive system than the CF-CT and the Brody-II scoring systems for more advanced disease in older patients. The PRAGMA-CF %Disease subscore was the most striking indicator

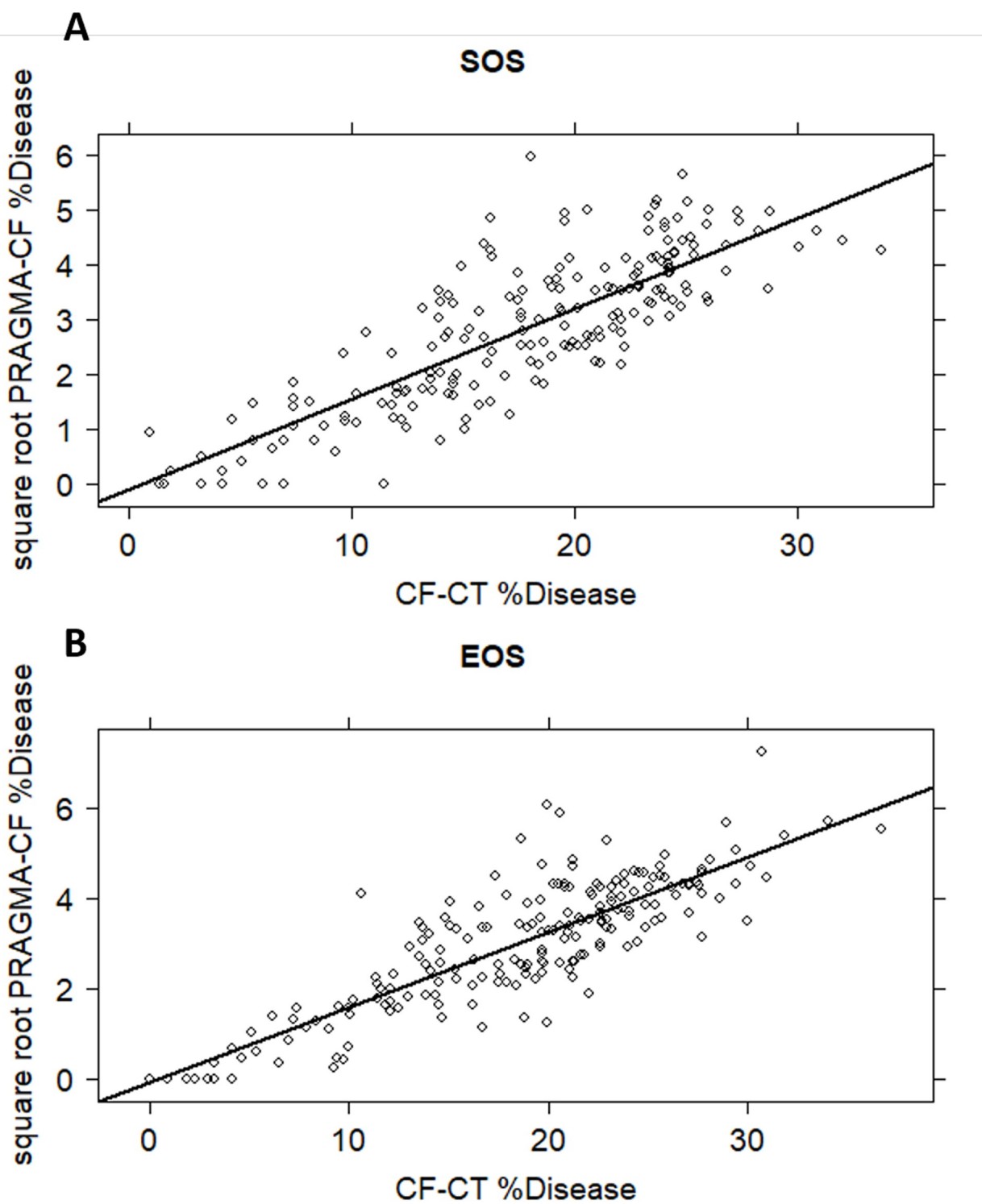

**Fig 3.** SOS (A) and EOS (B) CT scan image analysis results of %Disease subscore: CF-CT score vs. PRAGMA-CF method. This figure shows CF-CT %Disease scored by observer 1 plotted against PRAGMA-CF %Disease scored by observer 2 at the start of study (SOS, A) and at the end of study (EOS, B). The black line represents the regression line. The correlation between PRAGMA-CF and CF-CT for %Disease at SOS and EOS is 0.74 (p = <0.001).

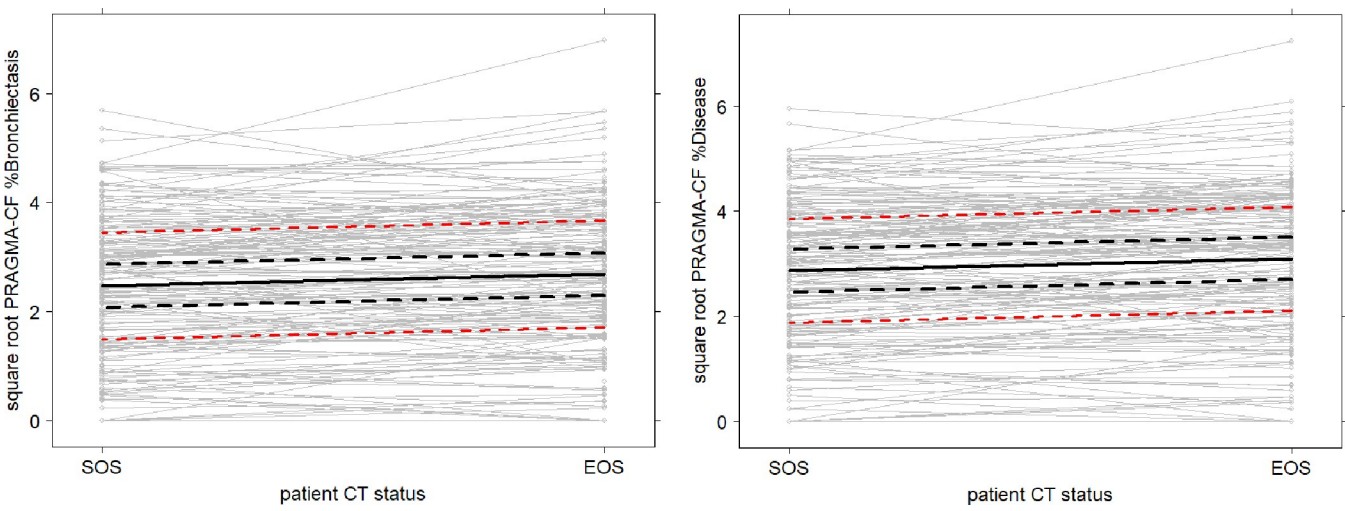

**Fig 4.** Change of PRAGMA-CF %BE (A) and %Disease (B) from EOS to SOS per patient. These plots show the change in PRAGMA-CF %BE (A) and PRAGMA-CF %Disease (B) from start of study (SOS) to end of study (EOS). Grey open circles represent the measurements of individual patients and the grey lines are plotted between the measurements of a single patient at SOS and EOS time-point. Black solid line represents the change obtained by the linear mixed model for patients with ataluren and tobramycin treatment. Note that for other subgroups the lines would not be statistically different as there was no significant impact for any of the other investigated confounders. The dashed black lines represent the confidence interval, and the dashed red lines the prediction interval. Note that for PRAGMA-CF %Disease values are presented in the square root scale—which is the scale that was used to analyse the data as the errors for this subscore were not normal distributed. Furthermore, note that due to the large number of patients included in the figure the grey open circles might appear filled.

of disease progression. This observation adds to the concept that %Disease subscore is an additional valuable tool to monitor changes in lung disease as it contains all relevant changes related to airways disease being %Bronchiectasis, %AWT, and %Mucus Plugging.

**Table 3. Results of the linear mixed effects models for the outcome PRAGMA-CF %Mucus plugging and % Disease.**

**PRAGMA-CF %Mucus Plugging**

|  | Value | Standard Error | p-value |
|---|---|---|---|
| (Intercept) | 1.114 | 0.098 | <0.0001 |
| patient CT status (EOS) | 0.175 | 0.080 | 0.0292 |
| Treatment group (placebo) | 0.124 | 0.132 | 0.3501 |
| Tobramycin treatment (Yes) | 0.226 | 0.154 | 0.1435 |
| patient CT status (EOS): Treatment group (placebo) | -0.129 | 0.099 | 0.1910 |
| patient CT status (EOS): Tobramycin treatment (Yes) | -0.116 | 0.100 | 0.2478 |
| Treatment group (placebo): Tobramycin treatment (Yes) | -0.136 | 0.204 | 0.5048 |

**PRAGMA-CF %Disease**

|  | Value | Standard Error | p-value |
|---|---|---|---|
| (Intercept) | 2.839 | 0.161 | <0.0001 |
| patient CT status (EOS) | 0.212 | 0.079 | 0.0079 |
| Treatment group (placebo) | -0.046 | 0.205 | 0.8240 |
| Tobramycin treatment (Yes) | 0.024 | 0.259 | 0.9251 |
| patient CT status (EOS): Treatment group (placebo) | -0.075 | 0.098 | 0.4427 |
| patient CT status (EOS): Tobramycin treatment (Yes) | 0.020 | 0.099 | 0.8375 |
| Treatment group (placebo): Tobramycin treatment (Yes) | 0.156 | 0.358 | 0.6633 |

Confounders included in the models were: Time: SOS/EOS, Tobramycin treatment: yes/no and Treatment group: ataluren/placebo.

As we showed a highly significant correlation between PRAGMA-CF and CF-CT subscores at SOS and at EOS we would also have expected a significant increase of CF-CT subscores as well as CF-CT is considered to be more sensitive compared to the Brody-II system used for the first ataluren study by PTC. The CF-CT scoring system is based on the Brody-II scoring system which includes standardized training of observers by using an extensive instruction module with clear definitions, reference images, and training sets. The first reason why the CF-CT scoring system did not detect progression is probably the coarse scales on which abnormalities like bronchiectasis are scored. For example, bronchiectasis can be scored for each lobe as absent or as occupying 0–33%, 33–66%, or >66% of the lobe. Relevant but subtler (less than 33%) structural changes in the ataluren study might therefore have been missed. The second reason why the CF-CT scoring system did not detect any significant progression in the ataluren study is probably related to the relative short study duration of 48 weeks. To date the CF-CT scoring method has been primarily used in cohort studies for which biennial CT scans were analysed over longer periods of time than 48 weeks [3, 12, 13]. Overall, our findings are in support of PRAGMA-CF as a sensitive outcome measure in clinical studies to monitor structural changes in CF lung disease.

The second important observation of our study is that we did not find a significant difference in change in CT scores between SOS and EOS for the placebo or ataluren groups in the intention to treat (ITT) analysis. This was also the case comparing treatment arms for all patients not receiving inhaled tobramycin treatment. Hence, our post hoc-analysis contrasts with the reported post hoc-analysis from the first phase 3 ataluren study that showed higher % predicted $FEV_1$ and fewer exacerbations in the ataluren group versus placebo in patients without inhaled tobramycin treatment.

In our analysis of 210 of the 238 study patients there was no longer significant improvement in $FEV_1$ from SOS to EOS in the ataluren treated patients not treated with tobramycin versus placebo. Our CT scan analysis is therefore in concordance with the negative findings of the second phase 3 ataluren RCT which did not observe a significant treatment effect on $FEV_1$ as outcome measure for ataluren treated patients versus placebo in patients naïve for tobramycin.

It has been well established in multiple observational studies that CT scores are more sensitive relative to $FEV_1$ to track progression of structural lung disease in CF patients [1, 3]. In today's CF care patients, using standard of treatment, have only little loss of functional outcomes such as $FEV_1$ over one year [14]. For this reason, it has become important to include more sensitive chest CT outcome measures for RCTs of drugs that aim to reduce structural damage of the lungs. Hence, our results favour a multi-modality approach including functional and structural outcomes to understand whether a drug is effective. The addition of the PRAGMA-CF scan analyses, being a more sensitive CT scan measure, could have been valuable in the interpretation of the post hoc subgroup analysis and for the decision to run a further phase 3 study.

Substantial heterogeneity of structural changes in the lungs was observed in these 210 nmCF patients at baseline as is shown in Fig 2. A similar heterogeneity disease spectrum was observed in patients with moderate to end stage lung disease [6, 15]. In end stage lung disease the median %Disease was 29% with a range between 18–42%. Nonsense mutations are considered severe mutations which is confirmed by the severity of the structural changes that we observed in the ataluren population with a mean PRAGMA-CF % of 9.8% and values as high as 32%. The most important volume component of %Disease is %Bronchiectasis followed by %Mucus Plugging. The contribution of %AWT is small in contrast to early CF lung disease where this is the most frequent abnormality found [16]. This is the consequence of the hierarchical scoring order of the PRAGMA-CF scoring method. An airway that shows bronchiectasis or mucus plugging mostly will have a thickened wall but the first option for scoring is

bronchiectasis, next is mucus plugging, and only when both are not apparent, is airway wall thickness assessed. Hence, this approach in patients with more advanced disease will result in an underestimation of airway wall thickness assessment. As %Disease includes all three relevant airway components it is not influenced by this hierarchical order.

Unfortunately, for this study we were not able to assess %Trapped Air as only 5% of patients had a volumetric expiratory scan which is needed for the accurate analysis of %Trapped Air [17].

For the development of more personalized treatments we suggest to include a chest CT scan in RCT at SOS to phenotype structural abnormalities of the study population and to improve our understanding of the response to treatment [18]. For example, PRAGMA-CF % Disease could be of use to predict the effect of an interventional drug on the clearance of mucus or airway wall thickness. Furthermore, the SOS CT scan gives us insight in the volume of structural changes that might still be reversible.

The results of this chest CT image analysis provides some guidance on the numbers of patients needed in a study that includes chest CT outcomes as it provides for the first time the 1-year natural history of CT change in CF [7, 8]. The significant increase of %Disease from SOS to EOS, observed in the current study, is largely the result of progression of %Mucus Plugging. The development of bronchiectasis is in general considered to be slow. It has been estimated that the ideal study duration for a disease modifying drug using %Bronchiectasis as outcome measure would be 2 years [19]. This PTC study supports the idea that for %Bronchiectasis a study duration of longer than 1 year is needed. Disease modifying drugs at best can stop progression of more advanced bronchiectasis, but is unlikely to reverse such bronchiectasis. This is not the case for PRAGMA-CF %Mucus Plugging which theoretically can be reversed for which reason a one-year study or even a shorter study duration of 3 to 6 months might be sufficient to measure its effect by CT scan analysis [20]. As PRAGMA-CF %Disease captures all changes related to airways disease it is a score that most likely can be improved over time. In addition %Disease would require only 110 subjects per study arm assuming a clinically relevant reduction of 50% in the active treatment arm versus placebo. This power can be substantially improved with standardizing CT scanning protocols, reconstruction techniques, and lung volume level during acquisition in clinical studies [18, 19, 21, 22]. This contributes importantly to the optimization of the image properties for image analysis balancing between radiation dose and image quality. This standardization is also important for reducing the number of CT scans that cannot be scored due to movement artefacts which was 7% for the PTC study. Furthermore, for PRAGMA-CF its sensitivity can be improved once it is automated. In addition, other promising automated image analysis systems are in development that are likely to be more sensitive to detect changes of airway dimensions [23]. For the development of such systems the PTC dataset is of great value for validation. Clearly for our power considerations one should keep in mind that we extrapolate from an observational trial scenario in mostly adult nmCF patients of a drug that was not effective to an interventional trial with a drug with a potential positive treatment effect. Ongoing studies in preschool children that are using chest CT as primary outcome measure will generate further evidence on the sensitivity of chest CT related outcomes in younger CF patients [18, 21].

In conclusion, progression of structural lung disease was observed in the 48 weeks ataluren nmCF RCT study cohort using PRAGMA-CF subscores. Importantly, in a post-hoc subgroup analysis of patients not treated with inhaled tobramycin, there was no significant treatment effect in ataluren treated patients on CT scan analysis outcomes using the PRAGMA-CF score. This analysis could have provided valuable additional data in assessing the validity of the positive treatment effect seen in $FEV_1$ and exacerbation. We also show that a more extensive analysis of this unique set of chest CT scans of the ataluren study illustrates the important role that

chest CT scan related outcomes can play in evaluating the effect of disease modifying drugs on CF lung disease in an RCT. These findings support the inclusion of chest CT scans in RCTs that aim to slow down progression or even reverse structural lung changes such as airway wall thickening related to CF lung disease.

## Supporting information

**S1 File.**
(DOCX)

**S1 Data.**
(XLSX)

## Author Contributions

**Conceptualization:** Harm A. W. M. Tiddens, Mariette Kemner-van de Corput.

**Data curation:** Harm A. W. M. Tiddens, Eleni-Rosalina Andrinopoulou, Joe McIntosh, J. Stuart Elborn, Eitan Kerem, Nynke Bouma, Jochem Bosch, Mariette Kemner-van de Corput.

**Investigation:** Nynke Bouma, Jochem Bosch.

**Methodology:** Harm A. W. M. Tiddens, Eleni-Rosalina Andrinopoulou, Joe McIntosh, J. Stuart Elborn, Eitan Kerem, Mariette Kemner-van de Corput.

**Supervision:** Harm A. W. M. Tiddens, Mariette Kemner-van de Corput.

**Writing – original draft:** Harm A. W. M. Tiddens, Eleni-Rosalina Andrinopoulou, Joe McIntosh, J. Stuart Elborn, Eitan Kerem, Nynke Bouma, Jochem Bosch, Mariette Kemner-van de Corput.

**Writing – review & editing:** Harm A. W. M. Tiddens, Eleni-Rosalina Andrinopoulou, Joe McIntosh, J. Stuart Elborn, Eitan Kerem, Mariette Kemner-van de Corput.

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
