## [Decision Letter · Decision Letter 0]

16 Jun 2020

PONE-D-20-14479

Chest computed tomography outcomes in a randomized clinical trial in cystic fibrosis:

Lessons learned from the first ataluren phase 3 study

PLOS ONE

Dear Dr. Tiddens,

Thank you for submitting your manuscript to PLOS ONE. After careful consideration, we feel that it has merit but does not fully meet PLOS ONE’s publication criteria as it currently stands. Therefore, we invite you to submit a revised version of the manuscript that addresses the points raised during the review process.

This is an interesting and well-designed study. From the reviewer comments, you can see that they found merit in your study. Please find the careful comments enclosed, especially for some clarifications regarding data presentation.

We look forward to receiving your revised manuscript.

Kind regards,

Sophie Yammine

Academic Editor

PLOS ONE

Journal Requirements:

2. We note that you have provided information about ethics board approval and informed patient consent in the Ethics Statement. We ask that you additionally include this information in your Methods section.

3. We note that you add 'No - some restrictions will apply' on our data-shareing policy. Please clarify the nature of these restrictions, ie. If due to ethical or legal reasons.

In your Data Availability statement, you have not specified where the minimal data set underlying the results described in your manuscript can be found. PLOS defines a study's minimal data set as the underlying data used to reach the conclusions drawn in the manuscript and any additional data required to replicate the reported study findings in their entirety. All PLOS journals require that the minimal data set be made fully available. For more information about our data policy, please see http://journals.plos.org/plosone/s/data-availability.

This investigator initiated study was funded through an unconditional grant by PTC

Pharmaceutics, Inc. The corresponding author was given full access to all data that

was requested to PTC. Only data was within the scope of the study were requested.

The corresponding author had final responsibility for the decision to submit for

publication.            

5. Thank you for stating the following in the Competing Interests:

The authors have declared that no competing interests exist.

We note that one or more of the authors have an affiliation to the commercial funders of this research study : PTC Therapeutics Inc.

Reviewers' comments:

Reviewer's Responses to Questions

**Comments to the Author**

1. Is the manuscript technically sound, and do the data support the conclusions?

Reviewer #1: Yes

Reviewer #2: Yes

2. Has the statistical analysis been performed appropriately and rigorously? 

Reviewer #1: Yes

Reviewer #2: Yes

3. Have the authors made all data underlying the findings in their manuscript fully available?

Reviewer #1: Yes

Reviewer #2: No

4. Is the manuscript presented in an intelligible fashion and written in standard English?

Reviewer #1: Yes

Reviewer #2: Yes

5. Review Comments to the Author

Reviewer #1: This is an interesting and well-designed study that re-evaluated the CT scans acquired during a phase 3 randomized controlled trial (RCT) with two recent scoring systems that have been developed after the completion of that study.

I have only minor suggestions for the authors as detailed below to improve clarity and presentation.

Introduction:

1) Page 3, line 63. After detailing the primary and secondary endpoints, I would add a sentence to introduce tertiary endpoints: “The remainder of the endpoints presented were tertiary or exploratory. Chest CT was included ...”

2) Line 63. “Chest CT” might be better replaced with “Change in lung CT scores”

Methods:

3) Study population. Indicate that inclusion and exclusion criteria of the first PTC phase III study

are detailed in the supplement.

4) Chest CT scan. Lines 121-125. Figure 1 should be described in a specific paragraph at the beginning of the Results section. Please, avoid to cite figure 1 multiple times in the methods and results section, as it is confusing.

5) CT Scoring. Lines 128-129. Same as comment 4.

6) CF-CT scoring system requires more details: which is the score range for each subscore? How much time requires the scoring of each CT scan for both the CF-CT and the PRAGMA-CF scoring systems?

7) Lines 152-155. See comment 4.

8) Statistics. Could the authors re-order this paragraph according to the order of the results for clarity? (first ù intra-observer reliability, next correlation analysis, …

9) Line 176. Replace “of” with “between”

10) Line 188-189. Bland-Altman figures had to be cited in the results section, not here. Here cite only the method.

Results:

11) Start the results section with a paragraph describing the flow chart reported in Figure 1. Also, sorry but I don’t understand the flowchart:

a) At lines 210-211 you reported 211 patients (27 patients with no CT scans), while in the flowchart you reported 210 patients and 28 patients with no CTscan. Could you check the numbers?

b) In the flowchart, indicate that of the 238 patients, 120 were ataluren and 118 placebo?

c) 9 CT scans were incomplete: what do you intend? Also, 20 CT scans were not suitable for image analysis: why? (scanner parameters, artifacts…?)

d) 391 CT scans: (195 SOS+195 EOS?)

e) 195 CT scans: from patients trated with placebo (98 placebo SOS + 98 placebo EOS?)

f) 196 CT scans: from patients treated with ataluren (97 ataluren SOS+ 97 ataluren EOS?)

12) Table 1 (n=207?) Weren’t 211?

13) Lines 244-245. “The mean CF-CT % disease score for patients on ataluren was 17.65…” add “at SOS, and increased to 18.14…(p-value)” Report the p-value even if not significant.

14) Lines 245-246. Add mean PRAGMA-CF %disease at EOS and the p-value.

15) Table 2. Add column titles “method” and “subscore”.

16) Line 277. “subscores” might be substituted with: “%disease score”

17) Line 284. “S Fig 1A and 1B”, but you are referring to “S Fig 2A and 2B”. Please adjust the order of the supplemental figures according to their appearance in the main text.

18) Lines 282-288. “Pearson correlation=…, regression analysis…” might be better substituted by (r=…, p<0.001).

19) Line 287. “S Fig. 3A, 3B and 3C” but you are referring to S Fig 1A, 1B and 1C. Also, Bland Altman plot for % airway wall thickening is not reported.

20) Line 295 please rephrase as “r=0.74, p<0.0001”. Report n for the correlation analysis.

21) Line 298. Delete “significant” as % bronchiectasis shows a trend towards progression, but it is not significant.

22) Line 305. S Table 5A and 5B, but you refers to 4A and 4B.

Discussion:

23) Lines 360-361. Add the score range to the methods section.

24) Line 378. Remove “were”.

25) Line 380 In table 3 you have reported the results of the linear mixed effects models for the PRAGMA-CF subscores and not for FEV1

25) The discussion would benefit of a paragraph comparing the two scoring systems: training required by the radiologists, time to score each CT scan, effect of CT parameters and image artifacts on score.

26) What do you expect from the air trapping score? Wouldn’t be possible to quantify air trapping (low ventilation areas) only on inspiratory scans by thresholding images at -950 HU?

27) Could you comment on the additional dose related to the use of CT scoring system as outcome measure in a clinical trial?

Reviewer #2: In this article, Tiddens et al. reanalysed chest CT scans from patients with cystic fibrosis who participated in a phase 3 randomized control trial (RCT) of ataluren using newer CT scoring systems to determine if these would be more sensitive at detecting progression of disease. The original published results of the RCT did now show any improvement in FEV1, which was the primary endpoint, in patients who received the treatment and instead there was a decline in FEV1 % predicted in both groups. Other endpoints in this trial included chest CT scans measured at the start and at the end of study, where analysis with Brody II CT scores showed no difference between disease groups and no progression of disease at the end of the trial period. In the current study, the authors used two newer CT scoring systems and hypothesized that significant progression of structural lung disease can be observed in the study cohort using the newly developed CF-CT and PRAGMA-CF scoring systems. The authors provide a good rationale for conducting this study and the study is well-conducted. Therefore, my comments are mostly minor:

1. In the Statistics methods, it is simply stated that Linear mixed effects models were used for the outcomes. It would be helpful to state what the models aimed to achieve i.e. to show progression of CT outcomes, compare effects of treatment etc.

2. It would be good to briefly mention the results of the simulation study in the Results section, rather than in the Discussion, even if further detail is in the supplementary material. It should also be stated that power calculations were for PRAGMA outcomes.

3. In the first paragraph of the results section, the authors discuss analysis completed in January 2017 and results remaining unchanged in 2018 after obtaining the full PTC study database. It is unclear of the relevance of this information. Is the current analysis the results of the full study database?

4. The results for effect of treatment are not adequately described in the Results, but only shown in the Table 3. Since this was one of the aims of the study it should be highlighted in the Results section. Similarly, the results for Tobramycin warrants mention.

5. In the methods of the Linear mixed effects model it’s stated that FEV1%predicted is included in the model however the results for this are not shown.

5. Lines 238 to 240 of the Results section, the term ‘merged’ is used twice in the sentence.

6. Abbreviations such as CF and FEV1 are not defined on first use.

6. PLOS authors have the option to publish the peer review history of their article (what does this mean?). If published, this will include your full peer review and any attached files.

Reviewer #1: No

Reviewer #2: No

---

## [Author Response · Author response to Decision Letter 0]

17 Sep 2020

• As indicated we have made further changes to meets PLOS ONE's style requirements, including those for file naming.

2. We note that you have provided information about ethics board approval and informed patient consent in the Ethics Statement. We ask that you additionally include this information in your Methods section.

• As indicated we have now included in the methods section the same information about ethics board approval and informed patient consent as in our Ethics Statement. 

 3. We note that you add 'No - some restrictions will apply' on our data-sharing policy. Please clarify the nature of these restrictions, ie. If due to ethical or legal reasons.

In your Data Availability statement, you have not specified where the minimal data set underlying the results described in your manuscript can be found. PLOS defines a study's minimal data set as the underlying data used to reach the conclusions drawn in the manuscript and any additional data required to replicate the reported study findings in their entirety. All PLOS journals require that the minimal data set be made fully available. For more information about our data policy, please see http://journals.plos.org/plosone/s/data-availability.

• As indicated we have now supplied underlying data used to reach the conclusions drawn in the manuscript and that is required to replicate the reported study findings in their entirety. A locked exel spreadsheet that contains all image analysis results of all CTs is now uploaded as part of the supplemental material. Upon request a code can be given to unlock the database. Access to the annotated CTs can be given within the Erasmus MC IT environment for legal / privacy reasons. Furthermore, depending on the question written permission by PTC for access to the CTs might be needed as PTS pharmaceuticals is the owner of the CT dataset.

• As indicated we have followed the recommendations above and have added A locked exel spreadsheet that contains all image analysis results of all CTs is now uploaded as part of the supplemental material.

• We have now shared the exell datasheet that contain the image analysis data of our CT analysis data on which the statistical analysis has been executed as defined in our methods section. 

This investigator initiated study was funded through an unconditional grant by PTC

Pharmaceutics, Inc. The corresponding author was given full access to all data that

was requested to PTC. Only data was within the scope of the study were requested.

The corresponding author had final responsibility for the decision to submit for

publication. 

• We have amended the funding and financial disclosure section as follows: 

o Funding This investigator initiated study was funded through an unconditional grant by PTC Pharmaceutics, Inc. The corresponding author was given full access to all data that was requested to PTC. Only data within the scope of the study were requested. The corresponding author had final responsibility for the decision to submit for publication. The funder provided support in the form of salaries for EA, NB, JB, and MKvdC. Funder had no role in the image analysis study design and analysis, decision to publish. 

o Conflicts of interest statement: JM worked at the time of the manuscript preparation at PTC and owns stock of PTC, he supplied us with relevant data when requested and he critically read the manuscript and gave relevant input where needed fulfilling all requirements for co-authorship. EK received a research grant through his hospital to perform the ataluren Phase III study and acted as Medical & Scientific Consulting Board member and received reimbursement for his time and for travel expenses. HT is director of the Erasmus MC LungAnalysis laboratory, he is a co-inventor of PRAGMA-CF which is patented. All financial aspects of the patent are handled by the Erasmus MC.

• See above

• As suggested we have added the funder statement and conflict of interest statement in the cover letter. 

5. Thank you for stating the following in the Competing Interests:

The authors have declared that no competing interests exist.

We note that one or more of the authors have an affiliation to the commercial funders of this research study: PTC Therapeutics Inc.

• As suggested we have adjusted the role of Joe McIntosh who at the time of the study was working at PTC. In addition we have reviewed and adjusted the author contributions. See our amended statements above.

• See our amended COI statement.

Within your Competing Interests Statement, please confirm that this commercial affiliation does not alter your adherence to all PLOS ONE policies on sharing data and materials by including the following statement: 

"This does not alter our adherence to PLOS ONE policies on sharing data and materials.” (as detailed online in our guide for authors http://journals.plos.org/plosone/s/competing-interests). If this adherence statement is not accurate and there are restrictions on sharing of data and/or materials, please state these. Please note that we cannot proceed with consideration of your article until this information has been declared.

• As discussed: Access to the annotated CTs can be given within the Erasmus MC IT environment for legal / privacy reasons. Furthermore, depending on the question written permission by PTC for access to the CTs might be needed as PTS pharmaceuticals is the owner of the CT dataset.

• We have as suggested included both an updated Funding Statement and Competing Interests Statement in the paper as well as in our cover letter

• In follow up on your comments we have scrutinized and updated all COI statements

• Changed according to guidelines  

Reviewers' comments:

Reviewer's Responses to Questions

Comments to the Author

1. Is the manuscript technically sound, and do the data support the conclusions?

Reviewer #1: Yes

Reviewer #2: Yes

2. Has the statistical analysis been performed appropriately and rigorously? 

Reviewer #1: Yes

Reviewer #2: Yes

3. Have the authors made all data underlying the findings in their manuscript fully available?

Reviewer #1: Yes

Reviewer #2: No 

• We have now uploaded a study’s minimal underlying data set as Supporting Information files.

4. Is the manuscript presented in an intelligible fashion and written in standard English?

Reviewer #1: Yes

Reviewer #2: Yes

5. Review Comments to the Author

Reviewer #1: This is an interesting and well-designed study that re-evaluated the CT scans acquired during a phase 3 randomized controlled trial (RCT) with two recent scoring systems that have been developed after the completion of that study.

I have only minor suggestions for the authors as detailed below to improve clarity and presentation.

Introduction:

1) Page 3, line 63. After detailing the primary and secondary endpoints, I would add a sentence to introduce tertiary endpoints: “The remainder of the endpoints presented were tertiary or exploratory. Chest CT was included ...” 

• We adjusted the sentence as suggested.

2) Line 63. “Chest CT” might be better replaced with “Change in lung CT scores”

We adjusted the sentence as suggested.

Methods:

3) Study population. Indicate that inclusion and exclusion criteria of the first PTC phase III study

are detailed in the supplement.

• We added this comment as suggested.

4) Chest CT scan. Lines 121-125. Figure 1 should be described in a specific paragraph at the beginning of the Results section. Please, avoid to cite figure 1 multiple times in the methods and results section, as it is confusing.

• We deleted the figure 1 citations from the methods section as suggested and mention Figure 1 only at the beginning of the results section.

5) CT Scoring. Lines 128-129. Same as comment 4.

• Changed as suggested, see above. 

6) CF-CT scoring system requires more details: which is the score range for each subscore? How much time requires the scoring of each CT scan for both the CF-CT and the PRAGMA-CF scoring systems?

• We added the following requested information to this section: Each subscore is scored for each lobe on the inspiratory scan as absent, occupying 0-33%, 33-66%, or >66% of the lobe, and next multipliers are applied. Resulting scores range from 0 to 72, 0 to 54, 0 to 36, and 0 to 54 for these 4 components, respectively. The maximal possible total CF-CT score summing these subscores is 216 points. For statistical analysis CF-CT subscores are expressed as % of maximal score. The CF-CT scoring system takes around 30 minutes per CT scan.

7) Lines 152-155. See comment 4.

• Changed as suggested, see above

8) Statistics. Could the authors re-order this paragraph according to the order of the results for clarity? (first ù intra-observer reliability, next correlation analysis, …

• Changed as suggested

9) Line 176. Replace “of” with “between”

• Changed as suggested

10) Line 188-189. Bland-Altman figures had to be cited in the results section, not here. Here cite only the method.

• Changed as suggested

Results:

11) Start the results section with a paragraph describing the flow chart reported in Figure 1. Also, sorry but I don’t understand the flowchart:

a) At lines 210-211 you reported 211 patients (27 patients with no CT scans), while in the flowchart you reported 210 patients and 28 patients with no CTscan. Could you check the numbers?

b) In the flowchart, indicate that of the 238 patients, 120 were ataluren and 118 placebo?

c) 9 CT scans were incomplete: what do you intend? Also, 20 CT scans were not suitable for image analysis: why? (scanner parameters, artifacts…?)

d) 391 CT scans: (195 SOS+195 EOS?)

e) 195 CT scans: from patients treated with placebo (98 placebo SOS + 98 placebo EOS?)

f) 196 CT scans: from patients treated with ataluren (97 ataluren SOS+ 97 ataluren EOS?)

• We thank the reviewer for pointing out these mistakes. In addition the way we mixed numbers of CT scans and number of patients was inconsistent and confusing. We checked the numbers and they are now correct. We made changes to the flow chart (Fig 1) and text to make this more clear. 

12) Table 1 (n=207?) Weren’t 211?

• Thank you for discovering this error. 211 is the correct number of patients from whom we were able to analyze 195 scans at baseline and 196 scans at follow up. 179 patients had both SOS and EOS, 16 patients had only a SOS and 17 EOS CT. We have now computed baseline characteristics only for 195 patients who had a SOS CT which we considered more correct. 

13) Lines 244-245. “The mean CF-CT % disease score for patients on ataluren was 17.65…” add “at SOS, and increased to 18.14…(p-value)” Report the p-value even if not significant.

• In this paragraph entitled ‘Spectrum of disease at SOS and EOS’ we focus on the wide distribution of scores and not on the change. Therefore we did add ‘at SOS’ but preferred not to include the EOS values and the p values of the difference. 

14) Lines 245-246. Add mean PRAGMA-CF %disease at EOS and the p-value.

• See answer on comment 13.

15) Table 2. Add column titles “method” and “subscore”.

• Changed as suggested

16) Line 277. “subscores” might be substituted with: “%disease score”

• Changed as suggested

17) Line 284. “S Fig 1A and 1B”, but you are referring to “S Fig 2A and 2B”. Please adjust the order of the supplemental figures according to their appearance in the main text.

• Sorry for this omission. We adjusted the order of the supplemental figures. 

18) Lines 282-288. “Pearson correlation=…, regression analysis…” might be better substituted by (r=…, p<0.001).

• We adjusted this notation as suggested. 

19) Line 287. “S Fig. 3A, 3B and 3C” but you are referring to S Fig 1A, 1B and 1C. Also, Bland Altman plot for % airway wall thickening is not reported.

• We excuse ourselves for this mix-up. We only showed the Bland Altman plots for the 3 outcomes that showed a significant change in the mixed models. We now clarify this in the text. 

20) Line 295 please rephrase as “r=0.74, p<0.0001”. Report n for the correlation analysis.

• We adjusted this notation as suggested. 

21) Line 298. Delete “significant” as % bronchiectasis shows a trend towards progression, but it is not significant.

• Deleted as suggested

22) Line 305. S Table 5A and 5B, but you refers to 4A and 4B.

• We excuse ourselves for this mix-up which thankfully was detected by the reviewer. 

Discussion:

23) Lines 360-361. Add the score range to the methods section.

• As suggested we added the scoring ranges to the methods section. 

24) Line 378. Remove “were”.

• Deleted as suggested

25) Line 380 In table 3 you have reported the results of the linear mixed effects models for the PRAGMA-CF subscores and not for FEV1

• Based on the comment by the reviewer we added the following sentences to the results section of the mixed models: ‘ FEV1 did not show a significant change from SOS to EOS. There was no significant impact for ataluren and tobramycin treatment at baseline and over time on the linear mixed models results for %Bronchiectasis, %Mucus Plugging, %Disease and FEV1.’ In addition ‘All the other linear mixed model results for CF-CT subscores and %PRAGMA-CF subscores and FEV1% are shown in S Table 4A and 4B’

25) The discussion would benefit of a paragraph comparing the two scoring systems: training required by the radiologists, time to score each CT scan, effect of CT parameters and image artifacts on score.

• We feel that it is beyond the scope of the manuscript to add more details on the scoring methods to the discussion section. We added information on the training for CF-CT and for PRAGMA-CF in the supplemental material section. We did not study the effect of CT parameters and image artifacts on the scores. However, we do mention in the discussion that 7% of CTs were excluded from analysis because of poor image quality. Standardization of CT protocols and lung volume will further improve the sensitivity of the scoring systems to track disease. 

26) What do you expect from the air trapping score? Wouldn’t be possible to quantify air trapping (low ventilation areas) only on inspiratory scans by thresholding images at -950 HU?

• On the inspiratory scan low attenuation regions due to hypoperfusion in CF might be detected using thresholding. However, a spirometer controlled CT acquisition protocol is needed to obtain reliable data. 

27) Could you comment on the additional dose related to the use of CT scoring system as outcome measure in a clinical trial?

• Routinely used low dose protocols can be used for scoring inspiratory scans and ultra-low dose for expiratory scans. This is risk benefit is discussed in reference 18 which we have now added to the standardization sections which we expanded as follows: ‘This power can be improved with standardizing CT scanning protocols, reconstruction techniques, and lung volume level during acquisition in clinical studies [18, 19, 21, 22]. This contributes importantly for the optimization of the image properties for image analysis balancing between radiation dose and image quality. This standardization is also important reducing the number of CT scans that cannot be scored due to movement artifacts which was 7% for the PTC study.’

 

Reviewer #2: In this article, Tiddens et al. reanalysed chest CT scans from patients with cystic fibrosis who participated in a phase 3 randomized control trial (RCT) of ataluren using newer CT scoring systems to determine if these would be more sensitive at detecting progression of disease. The original published results of the RCT did now show any improvement in FEV1, which was the primary endpoint, in patients who received the treatment and instead there was a decline in FEV1 % predicted in both groups. Other endpoints in this trial included chest CT scans measured at the start and at the end of study, where analysis with Brody II CT scores showed no difference between disease groups and no progression of disease at the end of the trial period. In the current study, the authors used two newer CT scoring systems and hypothesized that significant progression of structural lung disease can be observed in the study cohort using the newly developed CF-CT and PRAGMA-CF scoring systems. The authors provide a good rationale for conducting this study and the study is well-conducted. Therefore, my comments are mostly minor:

• We thank the reviewer for the time spend on our manuscript and for the comments which were helpful to improve the manuscript.

1. In the Statistics methods, it is simply stated that Linear mixed effects models were used for the outcomes. It would be helpful to state what the models aimed to achieve i.e. to show progression of CT outcomes, compare effects of treatment etc.

• As suggested we changed the sentence on mixed models to: Linear mixed effects models [10] were used to assess progression and compare effects of treatment for the following CT outcomes between SOS and EOS:

2. It would be good to briefly mention the results of the simulation study in the Results section, rather than in the Discussion, even if further detail is in the supplementary material. It should also be stated that power calculations were for PRAGMA outcomes.

• As suggested we now briefly included the key results of the simulation study for the PRAGMA-CF outcomes and the referral to Table 5A and 5B in the Results section and deleted these results from the discussion.

3. In the first paragraph of the results section, the authors discuss analysis completed in January 2017 and results remaining unchanged in 2018 after obtaining the full PTC study database. It is unclear of the relevance of this information. Is the current analysis the results of the full study database?

• We felt it relevant to mention this as it makes clear that we could not have been biased in our analysis by the results of the 2nd Phase III study. The reviewer is correct that the current analysis is based on the full study database which included for example the allocation to study arm and allowed us to investigate potential differences in outcomes between study arms. Bases on your question we have deleted this sentence. The section now reads as follows. ‘Our image analysis was completed and database locked by January 2017. March 2017, PTC communicated the the negative results of the consecutive phase 3 study performed to validate the post-hoc group analysis results in those patients not treated with inhaled tobramycin as tobramycin was hypothesized to interfere with ataluren ribosomal binding mechanism of action. In 2018 after we obtained the relevant sections of the PTC study database which allowed us to execute an in-depth analysis including comparing study arms.

4. The results for effect of treatment are not adequately described in the Results, but only shown in the Table 3. Since this was one of the aims of the study it should be highlighted in the Results section. Similarly, the results for Tobramycin warrants mention.

• We thank the reviewer for this useful comment. We focused the paper on our hypothesis ‘We hypothesized that significant progression of structural lung disease can be observed in the study cohort using CF-CT and PRAGMA-CF scoring systems.’ However, as was stated in the introduction: ‘We also did a post hoc analysis to investigate whether our more sensitive CT scan analysis would have given useful information to the effectiveness of ataluren in nmCF patients with or without tobramycin treatment.’ Hence we did not want to put to much emphasis on this post hoc analysis. Based on the comment by the reviewer we added the following sentences to the results section of the mixed models: ‘ FEV1 did not show a significant change from SOS to EOS. There was no significant impact for ataluren and tobramycin treatment at baseline and over time on the linear mixed models results for %Bronchiectasis, %Mucus Plugging, %Disease and FEV1.’ 

In the methods of the Linear mixed effects model it’s stated that FEV1%predicted is included in the model however the results for this are not shown.

• We have now added the following sentence to the results section of the linear mixed models: ‘FEV1 did not show a significant change from SOS to EOS. In addition we state that ‘All the other linear mixed model results for CF-CT subscores and %PRAGMA-CF subscores and FEV1% are shown in S Table 4A and 4B.’

5. Lines 238 to 240 of the Results section, the term ‘merged’ is used twice in the sentence.

• We deleted the first merged

6. Abbreviations such as CF and FEV1 are not defined on first use.

• We have introduce the abbreviations after first use. 

---

## [Decision Letter · Decision Letter 1]

6 Oct 2020

Chest computed tomography outcomes in a randomized clinical trial in cystic fibrosis:

Lessons learned from the first ataluren phase 3 study

PONE-D-20-14479R1

Dear Dr. Tiddens,

We’re pleased to inform you that your manuscript has been judged scientifically suitable for publication and will be formally accepted for publication once it meets all outstanding technical requirements.

Kind regards,

Sophie Yammine

Academic Editor

PLOS ONE

.

Reviewers' comments:

Reviewer's Responses to Questions

**Comments to the Author**

1. If the authors have adequately addressed your comments raised in a previous round of review and you feel that this manuscript is now acceptable for publication, you may indicate that here to bypass the “Comments to the Author” section, enter your conflict of interest statement in the “Confidential to Editor” section, and submit your "Accept" recommendation.

Reviewer #1: All comments have been addressed

Reviewer #2: All comments have been addressed

2. Is the manuscript technically sound, and do the data support the conclusions?

Reviewer #1: Yes

Reviewer #2: (No Response)

3. Has the statistical analysis been performed appropriately and rigorously? 

Reviewer #1: Yes

Reviewer #2: (No Response)

4. Have the authors made all data underlying the findings in their manuscript fully available?

Reviewer #1: Yes

Reviewer #2: (No Response)

5. Is the manuscript presented in an intelligible fashion and written in standard English?

Reviewer #1: Yes

Reviewer #2: (No Response)

6. Review Comments to the Author

Reviewer #1: (No Response)

Reviewer #2: (No Response)

7. PLOS authors have the option to publish the peer review history of their article (what does this mean?). If published, this will include your full peer review and any attached files.

Reviewer #1: No

Reviewer #2: No

---

## [Editor Report · Acceptance letter]

21 Oct 2020

PONE-D-20-14479R1 

Chest computed tomography outcomes in a randomized clinical trial in cystic fibrosis:
Lessons learned from the first ataluren phase 3 study 

Dear Dr. Tiddens:

I'm pleased to inform you that your manuscript has been deemed suitable for publication in PLOS ONE. Congratulations! Your manuscript is now with our production department. 

Kind regards, 

on behalf of

Dr. Sophie Yammine 

Academic Editor

PLOS ONE